# SRC Kinase in Glioblastoma: News from an Old Acquaintance

**DOI:** 10.3390/cancers12061558

**Published:** 2020-06-12

**Authors:** Claudia Cirotti, Claudia Contadini, Daniela Barilà

**Affiliations:** 1Department of Biology, University of Rome “Tor Vergata”, 00133 Rome, Italy; claudiacirotti89@gmail.com (C.C.); claudiacontadini@gmail.com (C.C.); 2Laboratory of Signal Transduction, IRCCS-Fondazione Santa Lucia, 00179 Rome, Italy

**Keywords:** glioblastoma, SRC kinase, receptor tyrosine kinase, tumor microenvironment, inflammation, metabolic reprogramming, kinase inhibitor, cancer therapy, therapy resistance

## Abstract

Glioblastoma multiforme (GBM) is one of the most recalcitrant brain tumors characterized by a tumor microenvironment (TME) that strongly supports GBM growth, aggressiveness, invasiveness, and resistance to therapy. Importantly, a common feature of GBM is the aberrant activation of receptor tyrosine kinases (RTKs) and of their downstream signaling cascade, including the non-receptor tyrosine kinase SRC. SRC is a central downstream intermediate of many RTKs, which triggers the phosphorylation of many substrates, therefore, promoting the regulation of a wide range of different pathways involved in cell survival, adhesion, proliferation, motility, and angiogenesis. In addition to the aforementioned pathways, SRC constitutive activity promotes and sustains inflammation and metabolic reprogramming concurring with TME development, therefore, actively sustaining tumor growth. Here, we aim to provide an updated picture of the molecular pathways that link SRC to these events in GBM. In addition, SRC targeting strategies are discussed in order to highlight strengths and weaknesses of SRC inhibitors in GBM management, focusing our attention on their potentialities in combination with conventional therapeutic approaches (i.e., temozolomide) to ameliorate therapy effectiveness.

## 1. Introduction

Glioblastoma multiforme (GBM) is the most common and lethal primary malignant cancer of the central nervous system (CNS). The median survival of GBM patients is 13–15 months, and the overall five-year survival remains extremely low at approximately 5% [1]. 

GBM was the first cancer type to be systematically studied by The Cancer Genome Atlas Research Network (TCGA) [2]. Genomic and transcriptomic analysis have identified the deregulation of TP53, RB1, and receptor tyrosine kinases (RTK)/Ras/PI3K pathway as very highly frequent, and have pointed to these signaling as a core requirement for GBM pathogenesis [3]. The aberrant activation of RTKs results in deregulation of their downstream signaling cascade which includes activation of MAPK and ERK pathways, as well as activation of non-receptor tyrosine kinases, such as SRC and ABL [4].

GBM malignancy is supported by the large heterogenicity of these tumors characterized by a variety of cancer cells that are continuously evolving and by an important tumor microenvironment (TME). Tumor and nontumor cells in the TME cooperate to promote the aberrant expression of a panel of inflammatory cytokines that trigger neoangiogenesis and evasion from the immune response, further enhancing tumor aggressiveness [5]. Moreover, the cooperation between tumor and nontumor cells in the TME ensures the adaptation of cancer cells, exploiting metabolic reprogramming required to support tumor growth [6]. GBM treatment includes maximum safe surgical resection, followed by radiotherapy and concomitant and maintenance temozolomide chemotherapy [7]. Temozolomide (TMZ) is an orally administrated alkylating agent that increases reactive oxygen species (ROS), causes DNA damage, and induces apoptotic cell death. Although partially beneficial, GBM usually achieves resistance to TMZ, which is responsible for the poor survival associated with these tumors [7]. The main resistance mechanisms are again associated with tumor heterogenicity and evolution, sustained inflammatory cytokines production, rewiring of the metabolism, and finally, with aberrant functionality of DNA damage response and DNA repair mechanisms [8,9,10,11]. 

The connections between the aberrant activation of RTKs signaling and the deregulation of the aforementioned abnormalities that contribute to GBM development and to its resistance to therapy have been only partially uncovered. Being the hyperactivation of SRC kinase a central node downstream the constitutive activation of RTKs, we aim here to discuss its possible significance in GBM. We review the molecular mechanisms that trigger SRC kinase activation in this context and focus our attention on how SRC activity can sustain GBM through the regulation of networks that control inflammation and metabolism. Finally, we discuss the significance and the concerns about SRC targeting as a valuable therapeutic strategy. 

## 2. Receptor Tyrosine Kinases

Genome-wide studies have identified the aberrant functionality of RTKs as a main feature of GBM [3] and among them, epidermal growth factor receptor (EGFR) is constitutively activated in about 57% of GBM. EGFR (ErbB1) is a receptor tyrosine kinase that belongs to the ErbB family and includes ErbB2 (HER2), ErbB3, and ErbB4. In normal conditions, EGFR is activated upon binding to its ligands such as epidermal growth factor (EGF). In GBM, this regulation is frequently lost and EGFR signaling is, therefore, upregulated through several mechanisms independently of ligands. EGFR expression can be amplified because of amplification of the chromosome 7 region that includes the *EGFR* gene, or it can be constitutively activated by specific mutations. The most common *EGFR* mutant, *EGFR variant (v)III*, has an extracellular domain truncation from exons 2 to 7 and is constitutively active in GBM independently of EGF [12]. 

In addition to EGFR deregulation, GBMs are often characterized by the aberrant activation of other RTKs. Indeed, overall at least one RTK was found altered in 67.3% of GBM and in some contexts, tumor heterogenicity included the simultaneous deregulation of two RTKs [12].

The prevalence of EGFR aberrant activation in GBM makes EGFR an excellent target. Indeed, to hit EGFR-oncogene addiction, both tyrosine kinase inhibitors and immune-mediated therapy approaches have been developed in recent years and clinical trials are ongoing, although, so far, results have been quite disappointing [12]. Tumor heterogenicity and evolution, as well as signaling redundancy, account for resistance or acquired resistance in most of the cases [12]. As RTKs share several downstream effectors that participate in its signaling, an alternative strategy would rely on the identification of druggable signals downstream RTKs activation that would represent valuable targets to significantly hit tumor growth. 

## 3. SRC Kinase

SRC was the first oncogene identified [13,14,15] and it is the prototype of a group of non-receptor tyrosine kinases, named the SRC family kinase (SFK), which display a conserved organization in the following domains: (1) a unique N-terminal region, named SRC homology (SH) 4 domain, that can be myristylated and can promote the interaction with the plasma membrane; (2) an SH3 domain that binds to proline-rich sequences, particularly those carrying the PxxP motif, and contributes to fold SRC in the inactive conformation via its interaction with the linker region between catalytic domain and SH2 domain; (3) an SH2 domain that drives SRC interaction with phosphorylated Y residues, resulting in intramolecular and intermolecular interactions; and (4) an SH1 domain, that is the catalytic kinase domain [16]. SRC structure revealed a complex of intramolecular interactions that ensure the modulation of its activity [16,17]. 

In cancer, SRC kinase activity is often aberrantly upregulated leading to the aberrant transduction of its signaling cascade, which can promote cell proliferation and migration and has been connected to tumor progression, neoangiogenesis, and metastatization [18,19]. It has been reported that SFKs activity was upregulated in GBM as compared with normal brain [20]. Moreover, high SRC activity has also been reported in a panel of GBM cancer cell lines [21]. Importantly, in GBM, *SRC* gene is not amplified nor mutated and its mRNA expression levels are not enhanced [2], indicating that the increased SRC activity in GBM, similarly to other tumors, relies mainly on the aberrant activation of RTKs and integrins [18,19].

Interestingly, the hyperactivation of SRC in GBM significantly contributes to sustain the rewiring of some of the main networks, including inflammation and metabolism, which contributes to the establishment of TME and tumor development. These evidences suggest that SRC targeting could be a valuable strategy to hit these pathways, and therefore impinge on tumor growth. 

## 4. Tumor Microenvironment and Inflammation

One of the main driving forces of GBM is connected to its TME which contributes to the large heterogenicity of this tumor and significantly supports tumor growth.

TME is constituted by all the noncancerous cells inside the tumor (stromal cells), including astrocytes, fibroblasts, immune cells, microglia/macrophages, endothelial cells, and pericytes. In addition, glioblastoma stem cells (GSCs) are also present, which are a small population of cells with invasive and proliferative properties, and give the tumor the ability to invade healthy brain tissue and contribute to therapy resistance [22]. Proteins and non-protein biomolecules (i.e., polysaccharides, hormones, nitric oxide, etc.) which are produced by all cell types in the TME, along with an altered extracellular matrix (ECM) and the interstitial fluid, contribute to the overall TME and to its ability to support tumor growth [5].

Importantly, a significant pool of proinflammatory cytokines, chemokines, and growth factors is secreted by cancer and noncancerous cells concurring to sustain the TME. This event is triggered by the activation of proinflammatory transcription factors, such as STAT3, NF-kB, and activator protein 1 (AP-1) [10,23,24]; overall it enhances proliferation, invasiveness, or stemness of GBM cells, and it promotes the suppression of tumor-specific immunity [25].

In particular, monocytic cells with different origin (i.e., tumor-associated macrophages or TAMs) are recruited into the TME together with the resident microglia by GSCs. All these cells are defined glioma-associated microglia/macrophages (GAMs) [5]. By secreting immune-modulatory factors, GAMs orchestrate the other immune cells that enter in the brain, thereby leading to chronic inflammation and facilitating tumor proliferation, survival, and invasion. These immune-modulatory factors include cytokines (such as TNF-α and TGF-β), chemokines (such as CX3CL1/Fractalkine, CCL2/MCP-1, and CCL5), and growth factors (including fibroblast growth factor (FGF-2) and granulocyte-monocyte colony stimulating factor (GM-CSF)) [25,26]. Among these, TGF-β released from TAMs, induces the matrix metalloproteinase expression (MMP-2, MMP-9) from the tumor, promoting extracellular matrix remodeling and degradation, thereby enhancing GSCs invasion. In addition, TGF-β secreted by GSCs, promotes the M2 immunosuppressive phenotype of GAMs, thereby inhibiting T cell proliferation and promoting tumor progression [25,26].

TME has a primary role in tumor progression [27,28] and resistance to therapy [10,29,30,31]. A first issue is related to its contribution to the high heterogenicity that impairs the selection of a strategy able to target all the different cell populations into the TME. Strikingly, standard current therapy targets predominantly non-stem cells, but it is basically ineffective against the small population of GSCs that is ultimately responsible for tumor relapse [22]. In addition, high levels of inflammatory factors, such as IL-6, IL-8, MCP-1, IL1β, and VEGF-α, have been identified both in the conditioned media of several GBM cell lines and in the microenvironment of clinical samples. Overall, these factors sustain the inflammation process inside the TME and significantly contribute to TMZ resistance in vitro [10,32,33]. 

### SRC Modulates Inflammation in the Tumor Microenvironment

Several studies have shown that the SRC signaling network has a key role in the regulation of the TME dynamism. SRC activation triggers downstream signaling through the RAS/MAPK and PI3K/AKT pathways, promoting tumor proliferation, survival, and invasion [34].

Furthermore, SRC influences the distribution of the protein components of focal adhesions, and it regulates the signal of integrins and the expression of ECM proteins [35,36]. Thus, alterations in SRC pathways lead to a reduction of cell–cell interaction and cell-ECM adhesion, supporting TME changes. 

Moreover, it has been demonstrated that SRC can modulate TME dynamism also orchestrating movement and infiltration of immune cells into tumor [37]. Interestingly, hyperactivation of SRC could occur both in cancer cells and in immune inflammatory cells due to the inflammatory cytokines released in the TME [38,39]. 

SRC hyperactivation, in turn, can promote activation or overexpression of proinflammatory transcription factors, such as STAT3, NF-kB, and AP-1 [40], thereby supporting this inflammatory TME [41]. In particular, tumor cells release chemokines (such as, SDF-1, MIP-1, MCP-1, MIP-2, etc.) that trigger the activation of SRC kinase in immune cells, which in turn release other cytokines (TNF-α, IL-1β, IL-6, etc.) reciprocally activating SRC in cancer cells through a positive feedback mechanism. Thus, SRC activation is driven by proinflammatory cytokines, and conversely, cytokine production is driven by SRC kinases in a cross talk between tumor and inflammation [37].

Inflammation in GBM is cancer-related and it is started by the tumor cells themselves [10]. However, the pathways involved in cytokines production in GBM cells, as well as the role of SRC in it, is still a complex and open question. 

In general, in tumors, oncogenic *RAS* mutation activates the production of cytokines, such as IL-6 and IL-8 [10,42,43]. Although *RAS* mutation is generally not involved in the GBM pathophysiology, EGFR amplification or mutant EGFRvIII expression can lead to RAS and PI3K/AKT pathways deregulation. Thus, EGFR deregulation in GBM drives a strong induction and secretion of IL-6 and IL-8, triggered, respectively, by AKT/SMAD5 signaling [44] and by the activity of transcription factors, such as AP-1 and NF-kB [10,45]. Interestingly, EGFR and mutant EGFRvIII promote NF-kB activity in GBM via AKT-dependent and independent mechanisms, further supporting cytokines production [46,47].

In addition, we recently observed that caspase-8 expression in GBM played a non-canonical function sustaining NF-κB activation, cytokine production (i.e., IL-6, IL-8, IL-1β, MCP-1, and VEGF-α), and tumor growth in vivo [33,48]. Importantly, we have also reported that the aberrant activation of SRC in GBM cell lines promoted caspase-8 phosphorylation on Y380, leading to neoplastic transformation [49]. On the basis of these data, we can speculate that SRC can promote NF-kB activity, and therefore TME, by phosphorylating caspase-8 on Y380. Future experiments should clarify whether SRC-dependent phosphorylation is required to support the caspase-8–NF-kB axis. 

## 5. Tumor Microenvironment and Metabolic Reprogramming

Cancer cells rely mainly on glycolysis for energy production. The Warburg effect, described by Otto Warburg in 1925, explains how cancer cells modify their metabolism to sustain high energy and biomolecules demand, by preferentially using glycolysis rather than oxidative phosphorylation to produce ATP [50]. In normal cells, glucose is metabolized via glycolysis to pyruvate to finally produce energy, whereas cancer cells undergo aerobic glycolysis, shifting pyruvate production from glucose to lactate generation, even in normoxic conditions. In so doing, they produce less ATP-2 vs. 36 molecules produced by oxidative phosphorylation. Despite this, the Warburg effect gives a selective advantage to highly proliferating GBM cells, supported by high availability of essential molecules for nucleotides, lipids, and nonessential amino acids biosynthesis. It has been demonstrated that GBM cells metabolize more than 90% of glucose via aerobic glycolysis to produce NAPDH and lactate [51]. Importantly, lactate secretion sustains the establishment of an acidic TME which strongly supports tumor growth, invasion, and vascularization and contributes to immune evasion [52,53]. Its release in the extracellular space, indeed, triggers the recruitment of myeloid cells around the tumor. Among them, macrophages and microglia acquire a tumorigenic immunosuppressive phenotype, supporting tumor growth, invasion, and angiogenesis [53,54].

Interestingly, stromal acidification has been shown to promote MMPs activation, thereby facilitating migration and infiltration of cancer cells. Indeed, glioma cells depend on lactate production to sustain TGF-β2-mediated expression of MMP-2 [55]. Moreover, the pharmacological targeting of lactate efflux from GBM cells strongly affects tumor survival and invasiveness, suggesting that it is worthwhile to explore therapeutic approaches aimed at shifting cancer cell metabolism and preventing lactate efflux [56]. However, metabolic targeting in GBM is particularly tricky, mainly because of the high heterogenicity of this tumor in which cancer cells may not be dependent only on the Warburg effect. Indeed, several new studies support the idea that mitochondrial oxidative phosphorylation is also maintained in cancer cells, thereby driving tumorigenesis and therapy resistance, at least in part [57,58]. As an example, GSCs metabolically differ from proliferating tumor cells preferring mitochondrial oxidative respiration [59]. Accordingly, the inhibition of oxidative phosphorylation in cancer cells can contribute to fight cancer cell survival [60,61,62]. In line with this heterogenicity, different deregulation of the pentose phosphate pathway (PPP), that physiologically produce NADPH and pentose phosphates, is observed in relation to different cells inside the tumor. The dependence on PPP in GBM, which is generally ensured by the upregulation of NAPDH-producing enzymes isocitrate dehydrogenase 1 (IDH1) and glucose-6-phosphate dehydrogenase (G6PDH) [63,64], is a survival mechanism that enables high rate proliferating glioma cells the possibility of continuously filling biosynthetic pathways [65] and it contributes to therapy resistance [63]. 

Importantly, the presence of hypoxic TME strongly affects the metabolic reprogramming and the switch between PPP and glycolysis [66]. Indeed, hypoxia promotes the upregulation of glycolytic enzymes, mainly due to the hypoxia inducible factors (HIFs) stabilization-sustaining migration and invasive phenotype to escape from hypoxic damage. On the contrary, glioma cells in the vascularized areas of the tumor are highly proliferating and predominantly sustained by PPP [67,68].

Overall, GBM tumor cells and TME nontumor cells influence each other and reciprocally exploit metabolic reprogramming to sustain cancer progression and survival. 

Altogether, the high dependency of GBM from the above described metabolic changes, suggests that targeting key pathways involved in reprogramming can help to fight cancer progression and, importantly, therapy resistance. A role for receptor and non-receptor tyrosine kinases in metabolic reprogramming has been widely suggested and studied [69,70]. 

### SRC Modulates Metabolic Reprogramming in the Tumor Microenvironment

Metabolic enzymes are finely regulated to supply the cellular energy demand. Post-translational modifications, first of all phosphorylation, strongly activate/inhibit enzymatic activity [71]. Remarkably, a correlation between SRC activation and the consequent phosphorylation of the hexokinase 1 (HK1)-rate limiting enzyme of glycolysis has been identified in several tumors including GBM [72]. Active SRC, indeed, interacts with both HK1 and the isoform hexokinase 2 (HK2) and phosphorylates them, respectively, on Y732 and Y686, therefore, activating the enzymes. This modification induces glycolysis and promotes cell growth, tumorigenesis, and invasion. Noteworthily, SRC-dependent HK1 and HK2 phosphorylation leads to the production of glucose-6-phosphate intermediate, stimulating both glycolysis and PPP [72,73]. Interestingly, SRC activity increases glucose uptake via HK1 and HK2 [72]. How HK1 and HK2 can affect glucose uptake is still unclear. Nevertheless, as early as 1987, the overexpression of oncogene *src* in 3T3 mouse fibroblasts was reported to be responsible for the overexpression of glucose transporters and, coherently, increased glucose uptake [74]. This suggested that SRC activity could also affect metabolism through modification of gene expression, probably activating alternative functions of well-known proteins. Indeed, evidences suggested that HK2 acted as a co-activator of NRF2 transcription factor in stressed glioma cells to modulate genes mostly involved in redox homeostasis [75]. This supports the critical role of HK2 in GBM tumorigenesis. Indeed, HK2 isoform, predominantly expressed in proliferating embryonic cells, is aberrantly re-expressed in GBM, where it drives aerobic glycolysis, uncontrolled proliferation, therapy resistance, and invasion [76]. 

Very interestingly, it has been demonstrated that SRC modulates glucose uptake and metabolism in breast cancer acting on a ERK1/2-dependent translation of MYC, which in turn promotes GLUT1 transcription [77]. Later on, the interplay between SRC and MYC has been described in several other cancers, such as melanoma, non-Hodgkin lymphoma, osteosarcoma, and lung cancer [78,79,80,81]. Although the link between SRC and MYC has not been uncovered in GBM, this observation was potentially and extremely interesting. Indeed, it has been well known that MYC gene is upregulated in several types of GBM [82,83] and that it sustained the self-renewal capacity and tumorigenicity of GSCs [84,85]. Moreover, MYC has been recently demonstrated to drive glycolysis and tumor addiction to metabolites necessary for glycolysis in GBM [86]. The aberrant activity of SRC in GBM along with the important role of MYC in GBM, led to speculate that the SRC-MYC axis could also have a role in this context and strengthen the hypothesis that targeting SRC could help to defeat cancer-associated adaptation, among which MYC-dependent metabolic reprogramming. 

Noteworthily, it has been demonstrated that SRC can affect glycolysis in cancer cells through a direct phosphorylation of pyruvate dehydrogenase (PDH), inhibiting its activity and driving the Warburg effect [87]. These evidences, on the one hand, highlighted the role of SRC in driving metabolic reprogramming and, on the other hand, clearly support SRC mitochondrial localization, as previously observed [88,89,90,91]. 

The role of tyrosine phosphorylation in mitochondrial functionality has been widely described [92,93]. Even in normal cells, it has been demonstrated that SRC sustains mitochondrial functionality, phosphorylating and activating complex I subunit NDUFB10 [94]. Remarkably, SRC phosphorylates EGFRvIII on Y845, and therefore promotes its translocation to the mitochondria. This event enhances mitochondrial functionality and oxidative phosphorylation, supporting tumor growth even in the presence of glucose deprivation and could partially contribute to the failure of EGFR-targeted therapies [95].

## 6. Targeting SRC in Glioblastoma

As pointed out before, the hyperactivation of SRC kinase in GBM is frequently observed as a consequence of RTKs aberrant signaling. This observation suggests that SRC can represent a nodal point shared by the aberrant activation of different RTKs, and therefore its targeting can overcome the resistance to therapy connected to the redundancy of multiple RTKs signaling pathways. In addition, its role in the inflammatory response and in the reprogramming of metabolism that overall sustains the tumor microenvironment, point to SRC kinase targeting as a valuable approach to ameliorate GBM treatment [96].

Here, we review different molecules and strategies that have been developed to target SRC kinase and discuss some issues connected to their nature. A major concern is related to the specificity of these compounds.

The majority of SRC tyrosine kinase inhibitors (STKIs) binds to the kinase adenosine triphosphate (ATP) pocket, which is conserved among kinases, thus, representing a potential cause of cross-reactivity [97]. In fact, STKIs display broad selectivity profiles which could result in not expected clinical applications, advantageous or not, depending on the tumor context [17,98]. As reviewed in [99] and more recently updated by the same author [100] some FDA-approved small molecule STKIs such as dasatinib, bosutinib, saracatinib, and ponatinib, were initially developed as SRC/ABL inhibitors and some of them are currently in use mostly for hematologic tumors, whereas no such promising effect, so far, has been observed for solid tumors and, in particular, for GBM. 

Below, we discuss state-of-the-art studies aimed at evaluating the role of SRC-targeting compounds in GBM treatment in vitro, in single and combined approaches (Table 1). 

### 6.1. Dasatinib

Dasatinib is a potent multikinase inhibitor of second generation approved, in June 2006, by the FDA for the treatment of chronic myeloid leukemia (CML) and Philadelphia positive subtype of acute lymphoblastic leukemia (Ph+ ALL) patients, given its high potential to inhibit BCR-ABL and turn off its aberrant signaling in this pathology [127]. One year later, studies demonstrated the ability of dasatinib to bind up to 30 different kinases, as shown by proteomic analysis, in contrast to other TKIs such as imatinib or nilotinib used for the same purposes [128,129]. Therefore, it showed high efficacy in the treatment of CML patients resistant to imatinib, acting against several kinases involved in the activation of the immune system [130,131,132]. Dasatinib inhibited SRC and other SFKs (i.e., FYN, LYN), and also c-kit, EPHA2, and PDGFR. For this reason, the possibility of using dasatinib for GBM treatment looked promising for turning off tyrosine kinases aberrant signaling and for sensitizing cells to canonical chemotherapeutic approaches using dasatinib alone or in combination with other drugs.

In this regard, in vitro and in vivo studies demonstrated that dasatinib affected migration, proliferation, and morphology of GBM cells through its inhibitory activity on SRC kinase [19,21,101,102]. In addition, it has been demonstrated that dasatinib treatment induced autophagic cell death mainly in combination with TMZ, thereby leading to a significant increase in the sensitivity to TMZ therapy, at least in glioma cells [103].

Interestingly, experiments using GBM cells resistant to TMZ, orthotopically xenografted in mice, have shown that dasatinib administration in combination with bevacizumab significantly reduced glioma cell invasion [104]. 

Unfortunately, despite these positive preclinical studies, the phase II clinical trial (NCT00892177) recently reported the failure of co-treatment with dasatinib and bevacizumab in patients with recurrent GBM, showing that the combination of these two drugs did not ameliorate the outcomes of GBM patients as compared with the single treatment with bevacizumab [105]. In addition, the phase II clinical trial (NCT00423735) reported a lack of activity of dasatinib against recurrent GBM [106,107]. The main problem was assumed to be the inefficient drug delivery beyond the blood–brain barrier (BBB). This hypothesis was confirmed by Agarwal et al., in 2019, who demonstrated that active efflux on the BBB limited dasatinib delivery to the brain tumor and, consequently, treatment efficacy [133]. Indeed, several trials have been set up to better evaluate the potentialities of dasatinib, however, their results are only partially available and are not conclusive (Table 2). 

### 6.2. PP2 

PP2 is a selective STKI described, for the first time in 1996, for its ability to inhibit some members of the SRC family (SRC, FYN, LCK), as the closely related PP1 [134]. Through its action on the SRC pathway, PP2 affects proliferation and migration of GBM cells. PP2 treatment prevents the SRC-dependent formation of CAS/Crk/RAC complexes responsible for cytoskeletal reorganization, thus, blocking the migration process [109]. Furthermore, PP2 also inhibits cell migration affecting SRC-mediated caveolin-1 phosphorylation [110].

Importantly, it has been demonstrated that PP2 increased radiosensitivity in U251 and T98G cells by suppressing the secretion of MMP2, which is well known to promote cell invasion and resistance to therapy [111,112]. In addition, PP2 compromised cell migration and invasion in vitro promoting E-cadherin expression and inhibiting VEGF and EphA2 expression. Interestingly, the combination of PP2 and the standard treatment of glioma (radiotherapy with TMZ) has been shown to affect tumor growth in nude mice [112]. 

### 6.3. SI221

SI221 is a novel selective pyrazolo [3,4-d] pyrimidine derivative SFK inhibitor, which is able to reduce cell migration and to promote cell death of GBM cells. S1221 showed more significant cytotoxic effect, higher metabolic stability, and a better potential to cross the BBB as compared with PP2 [113]. Studies to increase its solubility in water could further increase its potentialities [135].

### 6.4. Bosutinib

Bosutinib (SKI-606) is a third generation TKI that targets both SRC kinase and the oncogene ABL. Similar to dasatinib, it was approved by the FDA for therapeutic use for CML patients resistant to imatinib treatment due to ABL-BCR mutations [136]. Taylor et al. observed that although bosutinib had low brain penetration, it was still sufficient to achieve the drug concentration that inhibited SRC kinase activity in vitro. However, they demonstrated, in a phase II study, that bosutinib monotherapy was not associated with antitumor activity in recurrent GBM patients [114] (Table 2). The failure of bosutinib treatment efficacy in this trial was still unclear. It could have depended on the status of SRC activation in these patients with recurrent GBM (that is unknown) or it could have been due to the low drug delivery in the brain tumor, which remain the main problems of TKIs. 

### 6.5. Saracatinib

Saracatinib (AZD0530) is a dual inhibitor of both SFKs and ABL. Its antiproliferative effects have been observed in prostate cancer cell lines, where it suppressed the activation of SRC, thereby leading to G1 growth arrest [137]. In addition, saracatinib has been demonstrated to inhibit in vitro migration and invasion of breast tumor cells resistant to lapatinib treatment. These cells are characterized by a deregulated activation of several pathways, such as ERK, PI3K/AKT, and SRC signaling, which could induce their resistance to lapatinib [138].

A recent study by Liu et al. investigated the possible role of saracatinib to interfere with the aberrant STAT3 signaling and improve GBM treatment. GBM showed an overactivation of SRC pathway as described above, and consequently, of STAT3 protein that resulted in the activation of different events correlated with cell survival, growth, apoptosis, differentiation, and inflammation [139,140,141]. For this reason, Liu et al. investigated a novel strategy to hit GBM tumors based on the combination of lentiviral vectors to express siRNA targeting STAT3 and saracatinib. A combined therapy increased the apoptotic rate of GBM cells in vitro and decreased tumor growth in vivo more efficiently as compared with the single treatments. However, saracatinib in vivo was less effective, possibly because of drug metabolism and tumor drug resistance, highlighting how much more needs to be done to improve drug therapy [115].

### 6.6. SU6656

SU6656 is a small-molecule inhibitor of SRC kinase and of the other members of the SRC family (i.e., FYN, YES, and LYN) with a weaker activity towards the non-receptor tyrosine kinase ABL. SU6656 has been demonstrated to inhibit cell growth in U251 glioma cells and to reduce the invasiveness of glioma spheroid implanted in a three-dimensional collagen matrix due to the changes in actin dynamics [19,116]. In addition, in vitro and in vivo studies have demonstrated that SU6656 increased the tumor sensitivity to radiation inhibiting AKT phosphorylation. In this way, it promoted radiation-induced apoptosis and destruction of blood vessels inside the tumor, leading to a delay in tumor growth [117], suggesting that SU6656 antitumoral effect should be considered for GBM treatment. 

### 6.7. Ponatinib

Ponatinib was approved, in 2012, by the FDA for the treatment of resistant or intolerant CML and Ph+ ALL, as a third generation multikinase inhibitor targeting BCR-ABL, SRC, EGFR, PDGFR, FGFR, and VEGFR [142]. Importantly, in 2013, ponatinib was temporarily discontinued due to serious side effects, probably caused by the pan-activity of this TKI, and was then reintroduced only for selected patients [143].

It has been demonstrated that ponatinib has an antitumoral effect in U87 GBM cells, reducing cell viability, migration, invasion, and causing apoptotic cell death. Moreover, a reduction of tumor growth in vivo was observed, even if the use of heterotopic models limited this study, as also supported by previous evidences showing a critical difference in the ponatinib antitumoral effect between heterotopic and orthotopic models [118,119]. Recently, a phase II clinical trial with ponatinib was conducted on GBM patients who were resistant to bevacizumab. The idea behind the study was to try to overcome resistance therapy by targeting RTKs and non-RTKs with ponatinib. Unfortunately, the study closed early, i.e., at the first stage, due to its inefficacy [120] (Table 2). One possible explanation for this was the inability of the drug to reach the tumor, as previously mentioned [119].

### 6.8. Tyrosine Kinase Inhibitors in Glioblastoma: New Therapeutic Approaches

As previously discussed, GBM targeting is a complex issue. The main criticism that represent obstacles for a successful treatment of this tumor, is related to the location of the tumor inside the brain. Physiologically, the BBB protects the brain from exogenous insults, such as pathogens or toxins. The presence of tight junctions avoids the permeability of most of the circulating compounds inside the brain. The diffusion through a membrane only allows small lipophilic molecules, such as oxygen, while the presence of ATP-binding cassettes (ABC) transporters, such as P-glycoprotein (Pgp) and breast cancer resistance protein (BCRP), ensure the efflux of materials from the brain to the blood. This particular structure, although essential to ensure protection for the brain, is an enemy for therapies [144]. To overcome this issue, currently, one possible approach is to engineer nanoparticles that are able to cross the BBB mainly by using receptor-mediated endocytosis [145]. The difficulty behind the generation of nanocarriers able to deliver small molecules to the tumor site is, once again, related to GBM and BBB heterogenicity. Indeed, the composition of the BBB has to be carefully evaluated in order to successfully produce nanoparticles with the proper size and shape [146,147]. Until now, no promising results have been obtained and nanoparticles accumulation inside the brain appears to be very low [148], although several laboratories have been actively working to optimize and to develop new functional nanoparticles [147]. Regarding this issue, dasatinib has been recently conjugated to ultrasmall nanoparticles (cRGD-Das-NDCs) able to cross the BBB by binding α-integrins and reach the core of the tumor. Indeed, this strategy is promising as it overcomes the aforementioned problems for dasatinib distribution, allowing the efficient delivery of the drug and the selective inhibition of specific targets such as SRC and PDGFR [108]. 

Tumor development leads to an alteration of brain structures, in terms of vessels formation and tumor microenvironment reorganization. The BBB is also affected. Although not perfectly tight as in physiological conditions, the blood tumor barrier (BTB) has a very heterogenous structure characterized by both weaker tight junctions in some areas and identical structure of the BBB in some other areas. Although BBB properties in GBM are still debated, no univocal definition can be made. GBM heterogenicity is also reflected in the surrounding tissues, with some GBM areas displaying intact BBB and some others a partial disruption of this [149]. 

The high expression of ABC transporters in BBB/BTB represents a physical barrier that prevents directly targeting drugs to tumor cells, by actively pumping drugs outside the brain. Many FDA-approved drugs are direct targets of these transporters (Figure 1).

As an example, dasatinib is substrate of the Pgp and BCRP transporters; this results in the efficient efflux of the drug from the brain that basically does not reach the tumor site [133,150]. Indeed, the genetic inhibition of these transporters in BCRP/Pgp KO-mice allowed dasatinib to reach GBM inside the brain and to properly inhibit SRC kinase and its downstream pathways, resulting in reduced tumor growth and increased survival of KO mice as compared with the WT [133]. 

On the basis of the above studies, a valuable approach that could be beneficial to improve drug delivery to the tumor sites, is the targeting of the transporters [151,152]. Noteworthily, a new SRC kinase inhibitor has been developed, Si306, a pyrazolo [3,4-d] pyrimidines optimized between other similar compounds [153]. Si306 was chosen for its favorable absorption, distribution, metabolism, and excretion (ADME) and it has been demonstrated to have the ability to selectively and specifically inhibit SRC, and also the Pgp transporter. Indeed, differently from the commonly used dasatinib, whose treatment strongly affects Pgp expression, Si306 and the prodrug pro-Si306 do not increase the expression or the activity of the Pgp [121]. Interestingly, Si-306 has a potent antitumoral effect both in vitro and in vivo [122]. Importantly, orthotopic mice models have been used to show that this compound efficiently crossed the BBB, reaching the tumor site. Very interestingly, survival and behavior of mice were not affected and microscopic analysis of main metabolic organs, such as liver, kidney, and brain, did not show any signs of tissue alteration, thus, suggesting good tolerability and low toxicity for this compound [121]. 

Overall, these evidences make Si306 a good candidate to inhibit both SRC and Pgp transporters in anticancer therapy, suggesting that it would be worthwhile exploring combinatorial use of this drug with chemotherapeutic agents. Actually, a combinatorial approach has already been investigated demonstrating that pretreatment with Si306 sensitized U87 GBM cells to proton therapy [123]. Although this compound has great potential in GBM treatment, only preclinical studies are currently available. 

Another new compound developed as a non-ATP-competitive small molecule SRC kinase inhibitor is KX2-36. The promising efficacy of this drug resides in its dual mechanism of action. KX2-361 and its related compound KX-391 are indeed inhibitors of both SRC kinase and tubulin polymerization [124] without the collateral effects on neurite outgrowth and neuropathy often observed by canonical antitubulin treatments (i.e, vincristine, polymerization inhibitor or paclitaxel, depolymerization inhibitor). Although clinical trials with this compound are still missing, it has been clearly demonstrated that KX2-361 had good oral bioavailability and that it easily crossed the BBB, reached the tumor site, and delayed tumor progression, therefore, enhancing long-term survival in mice. Interestingly, KX2-361 had no effect in immunocompromised mice indicating that its efficacy relied also on the host immune system [125].

A very interesting new compound, atypical as compared with all the others reviewed here, is NEO100. Actually, it is not a proper SRC kinase inhibitor but it effectively turns off the SRC-dependent migratory phenotype of GBM invasive cells [126]. NEO100 is able to affect tumor growth and induce ER-stress dependent apoptotic cell death both in vitro and in vivo. Importantly, in vivo studies have demonstrated that daily intranasal treatment effectively reached tumor inside the brain, delayed tumor progression, induced apoptosis, and blocked tumor invasion, while no toxic effects were shown upon treatment. Even if the exact mechanism of action of NEO100 has not been elucidated yet, it has been observed that it strongly affected the expression of genes involved in cell migration and invasion. SRC activity is inhibited after NEO100 treatment, together with MAPK, AKT, and STAT3, whereas RhoA is activated [126]. Although this compound had a beneficial effect, the co-treatments with NEO100 and TMZ did not have any additive effect. Further investigations are needed to clarify the efficacy of this compound in clinic trials (Table 2).

## 7. Conclusions

The central role of SRC in the modulation of several pathways that sustain the interplay between inflammation and metabolic rewiring and overall contribute to TME in GBM (Figure 2), strongly supports its potentialities as a therapeutic target. Unfortunately, despite encouraging results obtained by several preclinical studies, so far, most of the clinical trials performed have failed.

The first issue for this defeat is connected to the incapacity of SRC-targeting drugs to reach the tumor site, because of the poor permeability of the BBB [114,133,149] (Figure 1). In addition to tumor location, intratumor heterogenicity and, in particular, the strong interplay between tumor cells and the TME, is a major concern [146]. To enhance the chances to successfully translate preclinical results into therapy, studies to ameliorate drugs permeability and to clarify the molecular bases and the networks triggered by SRC deregulation are, therefore, absolutely required. In particular, the characterization of SRC signaling, could lead to the identification of novel druggable targets to interfere with SRC’s ability to sustain the TME, and therefore ameliorate a tumor’s response to therapy. An alternative approach to enhance drug delivery could rely on a combination with a compound that targets the BBB functionality [144,145].

Finally, intertumoral heterogenicity in GBM patients is another fundamental issue that has to be considered [154], pointing to the requirement for studies aimed at improving patient stratification and identifying which GBM patients could indeed benefit from SRC targeting.

## Figures and Tables

**Figure 1 cancers-12-01558-f001:**
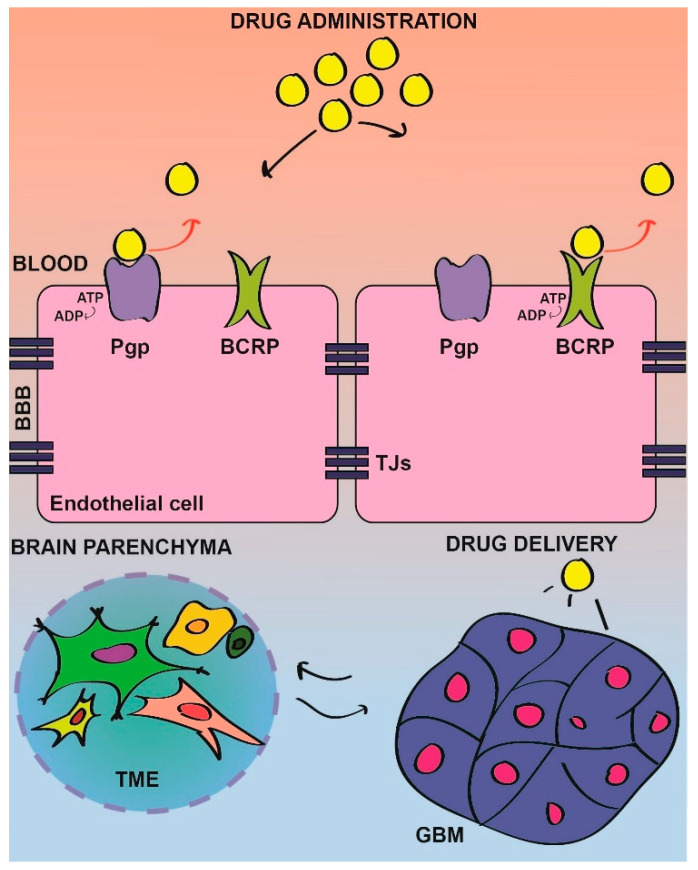
Schematic illustration of the drug delivery through the blood–brain barrier (BBB) in GBM. The BBB is characterized by ATP-binding cassettes (ABC) transporters, such as P-glycoprotein (Pgp) and breast cancer resistance protein (BCRP), located in the apical membrane of endothelial cells. Pgp and BCRP represent a physical barrier to the drug delivery into the tumor site, by actively pumping drugs outside of the brain into the blood flow (red arrows). Therefore, although a certain amount of drugs can still reach the brain parenchyma due to the heterogeneous permeability of the BBB in tumors-characterized by weaker tight junctions (TJs), in some areas, the required dose to obtain a therapeutic effect fails to reach the tumor site. TME, tumor microenviroment and GBM, glioblastoma.

**Figure 2 cancers-12-01558-f002:**
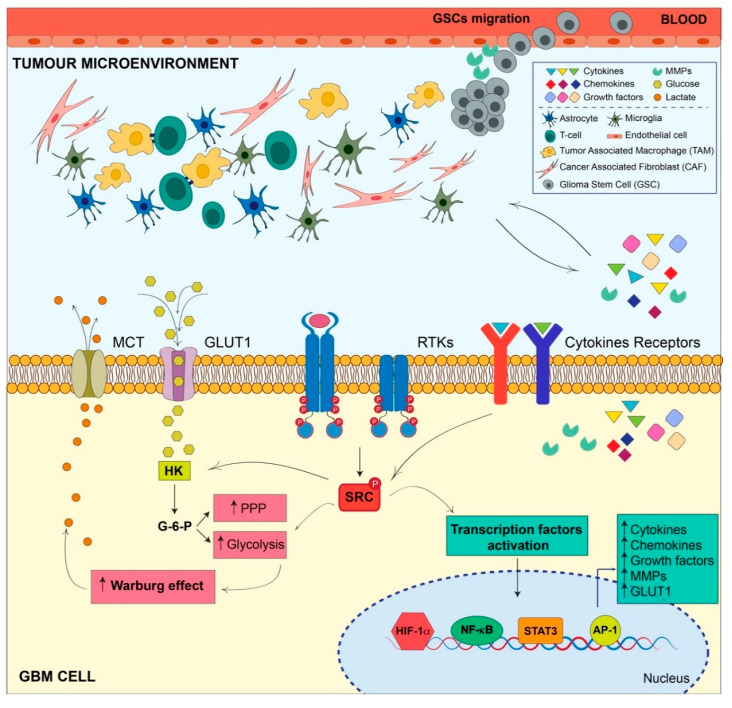
SRC constitutive activation modulates inflammation and metabolism concurring to tumor microenvironment sustainment in GBM. SRC constitutive activation in GBM is mainly caused by receptor tyrosine kinases (RTKs) aberrant signaling. SRC hyperactivation affects metabolism by inducing the activity of hexokinase (HK), responsible for the formation of glucose-6-phosphate (G-6-P) intermediate, feeding both the pentose phosphate pathway (PPP) and glycolysis. Importantly, SRC aberrant activity directly induces aerobic glycolysis (Warburg effect) causing lactate accumulation and its secretion in the extracellular matrix. SRC aberrant activity is also responsible for the stabilization and activation of transcription factors (i.e., NF-κB, STAT3, HIF1α, and AP-1) inducing the expression of proinflammatory cytokines, chemokines, growth factors, and matrix metalloproteases (MMPs). Massive production of pro-inflammatory molecules and growth factors, glucose deprivation, and lactate accumulation in the extracellular matrix concur with the formation of an important tumor microenvironment in which tumor and nontumor cells cooperate to sustain each other in a positive feedback loop concurring to GBM tumor aggressiveness and invasiveness.

**Table 1 cancers-12-01558-t001:** SRC tyrosine kinase inhibitors (STKIs) in glioblastoma treatment.

STKIs Name	Type of STKIs	Reference
DASATINIB	ATP-competitor	[21,101,102,103,104,105,106,107,108]
PP2	ATP-competitor	[101,109,110,111,112]
SI221	ATP-competitor	[113]
BOSUTINIB	ATP-competitor	[114]
SARACATINIB	ATP-competitor	[115]
SU6656	ATP-competitor	[116,117]
PONATINIB	ATP-competitor	[118,119,120]
Si306	ATP-competitor	[121,122,123]
KX2-361	Non-ATP-competitor	[124,125]
NEO100	Unknown	[126]

**Table 2 cancers-12-01558-t002:** Clinical trials for clioblastoma multiforme (GBM) treatment with STKIs.

STKIs Name	Clinical Trial (NCT Number)	Co-Treatment	Status	Reference
DASATINIB	NCT00423735	No	Completed	[106,107]
	NCT00892177	BEVACIZUMAB	Completed	[105]
	NCT00869401	IR/TMZ	Completed	n/a
	NCT00948389	LOMUSTINE	Terminated	n/a
	NCT00895960	IR/TMZ	Terminated	n/a
	NCT00609999	ERLOTINIB	Completed	n/a
	NCT00734864	TMZ	Withdrawn	n/a
BOSUTINIB	NCT01331291	No	Complete	[114]
PONATINIB	NCT02478164	No	Complete	[120]
NEO100	NCT02704858	No	Recruiting	n/a

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
