# Peer review of "SRC Kinase in Glioblastoma: News from an Old Acquaintance"

_cancers, 2020, doi:10.3390/cancers12061558_

Round 1

Reviewer 1 Report

The manuscript by Cirotti et al reviews recent insights into Src pathways supporting inflammation and metabolic reprogramming linked to Glioblastoma (GBM) growth. Furthermore, the authors discuss the pharmacological strategy to target Src and highlight strengths and weaknesses of SRC inhibitors in GBM therapeutic treatments.

In the complex the main text of the review article, giving some interesting informations, is carefully written. However, the reviewer thinks that the paper needs some improvements before publication.

In particular, the authors should explain, in SRC kinase section, the function of SH3 domain.

Moreover, the authors should ameliorate the quality of Figure 1 that doesn’t show and summarize the complexity of Src role in inflammation, metabolic reprogramming and tumour microenvironment regulation, during GBM growth. 

The authors should also add another figure depicting the blood–brain barrier and it’s role in affecting therapeutic strategies represented by tyrosine kinase inhibitors.

Minor:

Check typo.

Author Response

Reviewer 1

The manuscript by Cirotti et al reviews recent insights into Src pathways supporting inflammation and metabolic reprogramming linked to Glioblastoma (GBM) growth. Furthermore, the authors discuss the pharmacological strategy to target Src and highlight strengths and weaknesses of SRC inhibitors in GBM therapeutic treatments.

In the complex the main text of the review article, giving some interesting informations, is carefully written. However, the reviewer thinks that the paper needs some improvements before publication.

 We are glad that the Reviewer appreciate our review and we thank him/her for the suggestions and comments

In particular, the authors should explain, in SRC kinase section, the function of SH3 domain.

We have included this information in the Section 3, dedicated to SRC kinase (line 88-96, highlighted in red).

 Moreover, the authors should ameliorate the quality of Figure 1 that doesn’t show and summarize the complexity of SRC role in inflammation, metabolic reprogramming and tumour microenvironment regulation, during GBM growth.

We thank the reviewer for this suggestion. We replaced Figure 1 with a new figure, named Figure 2, that shows the major pathways that lead to SRC activation in GBM and the downstream signaling driving cytokine production, inflammation and tumor microenvironment regulation, as well as its ability to modulate metabolic reprogramming.

The old Figure 1 has been provided as graphical abstract as we believe it summarize the concept of the review.

The authors should also add another figure depicting the blood–brain barrier and it’s role in affecting therapeutic strategies represented by tyrosine kinase inhibitors.

We thank the reviewer for this suggestion and we included a new Figure 1 (Figure 1) that displays the role of the blood-brain barrier in the impairment of drug delivery. The depicted mechanisms affect several drugs including many tyrosine kinase inhibitors.  

Minor:

Check typo.

We carefully edited the manuscript.

Reviewer 2 Report

In this article, the authors summarized about the roles of SRC family kinases (SFKs) in maintenance of glioblastoma (GBM) biology and efficacy of SFKs as the therapeutic target of GBMs. The authors finally suggested simultaneous targeting of SFKs and blood brain barrier (BBB) might become novel therapeutic strategy against GBMs.

# Comments:

  • The authors should illustrate summary of the molecular network regulated by SFKs especially in GBMs by figures.

  • The authors should summarize clinical trials of SFKs inhibitors in GBM treatment by showing tables.

  • As the authors pointed out, elevated activation of SFKs in GBM tissue is considered as result of hyper activation of receptor tyrosine kinases (RTKs) or integrins. However, previous reports concluded targeting RTKs or integrins is considered as not effective in GBM therapy, and these evidences indirectly evoke targeting SFKs is not also valid in GBM therapy. Therefore, the authors should carefully consider and mention about this point in the text.

  • Because SFKs are housekeeping molecules in even normal tissue-derived cells, adverse effect is one of the critical problems in the therapies targeting SFKs. The authors expected dual targeting of both SFKs and BBB as novel therapeutic method against GBMs, however, in order to maximize concentration of SFKs inhibitors in GBM tissue, it is also necessary to use higher dose of SFKs inhibitors. This use of higher dose of SFKs inhibitors is considered to increase the probability of occurrence of adverse effects. Therefore, the authors should carefully address to this point in the text.

Author Response

Reviewer 2

In this article, the authors summarized about the roles of SRC family kinases (SFKs) in maintenance of glioblastoma (GBM) biology and efficacy of SFKs as the therapeutic target of GBMs. The authors finally suggested simultaneous targeting of SFKs and blood brain barrier (BBB) might become novel therapeutic strategy against GBMs.

# Comments:

The authors should illustrate summary of the molecular network regulated by SFKs especially in GBMs by figures.

 We thank the reviewer for this suggestion. This issue was pointed out also by the other Reviewer. We added a Figure in which we summarize the networks that we highlighted in the review pointing the attention to those pathways that lead to SRC activation in GBM and the downstream signaling driving cytokine production, inflammation and tumor microenvironment regulation, as well as its ability to modulate metabolic reprogramming. We included this figure as Figure 2 in the revised version. In addition, following the suggestion of Reviewer 1 we also included a picture that displays the major feature of the blood-brain barrier (BBB) that represents a major issue in the development of drugs that efficiently target GBM. This picture has been included as Figure 1.

The old Figure 1 has been provided as graphical abstract as we believe it summarize the concept of the review.

The authors should summarize clinical trials of SFKs inhibitors in GBM treatment by showing tables.

We agree with the reviewer that it is important to have a Table summarizing this topic. Indeed, we have already included in the review a Table, Table 2, in which we summarize the clinical trials that have been conducted or are ongoing to our knowledge, using drugs aimed to target SFKs in GBM. In this Table we provide several information:  the name of the SRC Tyrosine Kinase Inhibitor, the NCT number for the trial, whether the STKI was used in combination with other drugs of not, and the status of the Trial. We also provided the reference for a publication when available. We believe this is a quite complete information that helps the readers to get a summary of the topic. We can modify the table or add other information if the reviewer feels that something is missing.

As the authors pointed out, elevated activation of SFKs in GBM tissue is considered as result of hyper activation of receptor tyrosine kinases (RTKs) or integrins. However, previous reports concluded targeting RTKs or integrins is considered as not effective in GBM therapy, and these evidences indirectly evoke targeting SFKs is not also valid in GBM therapy. Therefore, the authors should carefully consider and mention about this point in the text.

We pointed out clearly that the elevated activation of SRC in GBM tissue is mainly due to the hyperactivation of RTKs or integrins. As pointed out by the Reviewers several drugs have been produced to hit different RTKs. This idea relies on the theory that cancer cells may become addicted to the aberrant activation of RTKs (Weinstein, 2002, Science) and indeed the administration of drugs to hit RTKs has been successfully introduced in the therapeutic approaches for many tumors (a well-known example is provided by the use of Trastuzumab in HER2 positive Breast Cancer Tumors). Regarding the failure of RTKs inhibitors in GBM therapy we believe that the reason is not linked to their ability to hit the RTK but more  to two major reasons that affect their efficacy in vivo:

  • GBM tumors are largely heterogenous tumors and a better stratification of patients is absolutely required in order to better classify the specific molecular abnormalities and design a more specific treatment by selecting the right drug combination
  • GBM tumors are generally difficult to target because of the Brain-Blood Barrier that is a physical obstacle to many drugs including several tyrosine kinase inhibitors

We believe therefore that the major challenge is to develop compounds that can access the tumor despite the BBB. We discussed this issue in the conclusions section. In addition, as pointed out by the Reviewer, the combination of those inhibitors with compounds that may dampen the functionality of the BBB, deserves more investigation and may provide some advantages. We added a sentence on this point.

We believe that, in agreement with these considerations, the failure of SRC kinase targeting drugs is mainly linked to the issues outlined above and may therefore be overcome by the production of new compounds that may efficiently penetrate the BBB. Finally, we would like to stress that the idea of targeting SRC has some advantages compared to RTKs targeting as SRC activation is downstream several RTKs, suggesting that its targeting may be beneficial to hit the signaling of different RTKs and may bypass the issue of RTKs-addiction switching that has been recognized as a major cause for the acquired resistance to therapy (Thorne et al., 2016, Neuro Oncol).

Because SFKs are housekeeping molecules in even normal tissue-derived cells, adverse effect is one of the critical problems in the therapies targeting SFKs. The authors expected dual targeting of both SFKs and BBB as novel therapeutic method against GBMs, however, in order to maximize concentration of SFKs inhibitors in GBM tissue, it is also necessary to use higher dose of SFKs inhibitors. This use of higher dose of SFKs inhibitors is considered to increase the probability of occurrence of adverse effects. Therefore, the authors should carefully address to this point in the text.

We thank the reviewer for this comment. As pointed out above the use of tyrosine kinase inhibitors relies on the oncogene addiction idea (Weinstein, 2002, Science): cancer cells become addicted to the aberrantly activated tyrosine kinases and therefore their targeting may be more effective on cancer cells compared to normal cells that are not addicted to that kinases (Swick et al., 2002, Trends Mol Med). In any case we are aware that the concentration of the drugs should always be as low as possible to minimize side effects and general toxicity. As mentioned before, the presence of the BBB is a general obstacle for any type of drug administration and in order to improve drug efficacy there are two main strategies: 1) design compounds optimized for their ability to penetrate the BBB; 2) as suggested by the Reviewer, combine those compounds that efficiently target tyrosine kinases in vitro with drugs that target the BBB in order to ameliorate their delivery.  A sentence on this issue has been included in the Conclusions Section.

Round 2

Reviewer 1 Report

In the legend of figure 1 the authors should write “tight junctions” next to the abbreviation TJs

Reviewer 2 Report

Reviewer 2

In this article, the authors summarized about the roles of SRC family kinases (SFKs) in maintenance of glioblastoma (GBM) biology and efficacy of SFKs as the therapeutic target of GBMs. The authors finally suggested simultaneous targeting of SFKs and blood brain barrier (BBB) might become novel therapeutic strategy against GBMs.

# Comments:

The authors should illustrate summary of the molecular network regulated by SFKs especially in GBMs by figures.

 We thank the reviewer for this suggestion. This issue was pointed out also by the other Reviewer. We added a Figure in which we summarize the networks that we highlighted in the review pointing the attention to those pathways that lead to SRC activation in GBM and the downstream signaling driving cytokine production, inflammation and tumor microenvironment regulation, as well as its ability to modulate metabolic reprogramming. We included this figure as Figure 2 in the revised version. In addition, following the suggestion of Reviewer 1 we also included a picture that displays the major feature of the blood-brain barrier (BBB) that represents a major issue in the development of drugs that efficiently target GBM. This picture has been included as Figure 1.

The old Figure 1 has been provided as graphical abstract as we believe it summarize the concept of the review.

I  satisfied with author’s responses.

The authors should summarize clinical trials of SFKs inhibitors in GBM treatment by showing tables.

We agree with the reviewer that it is important to have a Table summarizing this topic. Indeed, we have already included in the review a Table, Table 2, in which we summarize the clinical trials that have been conducted or are ongoing to our knowledge, using drugs aimed to target SFKs in GBM. In this Table we provide several information:  the name of the SRC Tyrosine Kinase Inhibitor, the NCT number for the trial, whether the STKI was used in combination with other drugs of not, and the status of the Trial. We also provided the reference for a publication when available. We believe this is a quite complete information that helps the readers to get a summary of the topic. We can modify the table or add other information if the reviewer feels that something is missing.

I’m very sorry I missed table 2 in your text. The information included in table 2 is sufficient, and that’s OK.

As the authors pointed out, elevated activation of SFKs in GBM tissue is considered as result of hyper activation of receptor tyrosine kinases (RTKs) or integrins. However, previous reports concluded targeting RTKs or integrins is considered as not effective in GBM therapy, and these evidences indirectly evoke targeting SFKs is not also valid in GBM therapy. Therefore, the authors should carefully consider and mention about this point in the text.

We pointed out clearly that the elevated activation of SRC in GBM tissue is mainly due to the hyperactivation of RTKs or integrins. As pointed out by the Reviewers several drugs have been produced to hit different RTKs. This idea relies on the theory that cancer cells may become addicted to the aberrant activation of RTKs (Weinstein, 2002, Science) and indeed the administration of drugs to hit RTKs has been successfully introduced in the therapeutic approaches for many tumors (a well-known example is provided by the use of Trastuzumab in HER2 positive Breast Cancer Tumors). Regarding the failure of RTKs inhibitors in GBM therapy we believe that the reason is not linked to their ability to hit the RTK but more  to two major reasons that affect their efficacy in vivo:

  • GBM tumors are largely heterogenous tumors and a better stratification of patients is absolutely required in order to better classify the specific molecular abnormalities and design a more specific treatment by selecting the right drug combination
  • GBM tumors are generally difficult to target because of the Brain-Blood Barrier that is a physical obstacle to many drugs including several tyrosine kinase inhibitors

We believe therefore that the major challenge is to develop compounds that can access the tumor despite the BBB. We discussed this issue in the conclusions section. In addition, as pointed out by the Reviewer, the combination of those inhibitors with compounds that may dampen the functionality of the BBB, deserves more investigation and may provide some advantages. We added a sentence on this point.

We believe that, in agreement with these considerations, the failure of SRC kinase targeting drugs is mainly linked to the issues outlined above and may therefore be overcome by the production of new compounds that may efficiently penetrate the BBB. Finally, we would like to stress that the idea of targeting SRC has some advantages compared to RTKs targeting as SRC activation is downstream several RTKs, suggesting that its targeting may be beneficial to hit the signaling of different RTKs and may bypass the issue of RTKs-addiction switching that has been recognized as a major cause for the acquired resistance to therapy (Thorne et al., 2016, Neuro Oncol).

I re-checked the text and confirmed above points were already mentioned in the text. I agreed with these author’s rebuttals.

Because SFKs are housekeeping molecules in even normal tissue-derived cells, adverse effect is one of the critical problems in the therapies targeting SFKs. The authors expected dual targeting of both SFKs and BBB as novel therapeutic method against GBMs, however, in order to maximize concentration of SFKs inhibitors in GBM tissue, it is also necessary to use higher dose of SFKs inhibitors. This use of higher dose of SFKs inhibitors is considered to increase the probability of occurrence of adverse effects. Therefore, the authors should carefully address to this point in the text.

We thank the reviewer for this comment. As pointed out above the use of tyrosine kinase inhibitors relies on the oncogene addiction idea (Weinstein, 2002, Science): cancer cells become addicted to the aberrantly activated tyrosine kinases and therefore their targeting may be more effective on cancer cells compared to normal cells that are not addicted to that kinases (Swick et al., 2002, Trends Mol Med). In any case we are aware that the concentration of the drugs should always be as low as possible to minimize side effects and general toxicity. As mentioned before, the presence of the BBB is a general obstacle for any type of drug administration and in order to improve drug efficacy there are two main strategies: 1) design compounds optimized for their ability to penetrate the BBB; 2) as suggested by the Reviewer, combine those compounds that efficiently target tyrosine kinases in vitro with drugs that target the BBB in order to ameliorate their delivery.  A sentence on this issue has been included in the Conclusions Section.

I realized author’s intention and confirmed the changes of article. It’s OK.